# On Star Selection Principles Theory

Ljubiša D. R. Kočinac

Faculty of Sciences and Mathematics, University of Niš, 18000 Niš, Serbia; lkocinac@gmail.com

**Abstract:** The aim of this paper is to review up-to-date recent results in the field of star selection principles, a rapidly growing area of topology, and to present a few new results.

**Keywords:** star selection principle; hyperspace; selective property (a); selective acc; set selection principles

## 1. Introduction

We use the usual topological notation and terminology as in [1]. Throughout the paper, unless otherwise stated, no separation axioms are assumed. The set of natural numbers is denoted by $\mathbb{N}$, and the first uncountable ordinal is denoted by $\omega_1$. If $\mathcal{F}$ is a family of subsets of a space $X$, $A \subset X$, $x \in X$, then

$$\mathrm{St}(A, \mathcal{F}) = \cup\{F \in \mathcal{F} : F \cap A \neq \varnothing\}$$

is the *star* of $A$ with respect to $\mathcal{F}$; $\mathrm{St}(\{x\}, \mathcal{F})$ is denoted by $\mathrm{St}(x, \mathcal{F})$.

The use of the star operator in topology has a long history. For more details, see [2] and [3]. Kočinac applied this operator in the field of selection principles and introduced and studied a number of selection principles using this operator [4–6]. The natural examples for star selection principles are selection principles in uniform spaces [6–8] and in topological (and more generally, topologized) groups [9,10]. The survey papers [6,7,11] contain a detailed exposition on star selection principles theory, an important subfield of selection principles theory. The number of researchers working in star selection principles theory is increasing. Several Ph.D. theses on topics in this field have been written in recent years ([12–19]), and many papers appeared in the literature (see, for example, [20–44]).

In this chapter, which can be viewed as a complement to the survey papers [7,11], we review up-to-date recent results in this field and present a few new results.

The basic idea in selection principles theory, one of the most dynamic areas of research in topology in the last 25–30 years (see, for instance, [45]), is:

There are two families of sets, say $\mathcal{A}$ and $\mathcal{B}$. For each sequence $A_1, A_2, \dots$ of elements of $\mathcal{A}$, one selects, by a prescribed procedure, $B_1 \subset A_1, B_2 \subset A_2, \dots$ so that from the chosen $B_1, B_2, \dots$, by an operation, we obtain an element of $\mathcal{B}$. In this way one can assign a property-selective $\mathcal{P}$ to (almost) each topological property $\mathcal{P}$.

Two classical (star) selection principles, which will be discussed here, are the Menger-type principle $S_{fin}(\mathcal{A}, \mathcal{B})$, where $B_n$s are finite, and the Rothberger-type principle $S_1(\mathcal{A}, \mathcal{B})$, where $|B_n| = 1$ for each $n$.

We use the following notations for collections of open covers of a space $X$:

1. $\mathcal{O}$ is the collection of open covers of $X$.

2. $\Omega$ (respectively, $\mathcal{K}$) is the collection of $\omega$-covers (respectively, $k$-covers) of $X$. An open cover $\mathcal{U}$ of $X$ is said to be an *$\omega$-cover* (respectively, a *$k$-cover* [46]) if each finite (respectively, compact) subset of $X$ is contained in a member of $\mathcal{U}$.

3. $\Gamma$ (respectively, $\Gamma_k$) is the collection of $\gamma$-covers (respectively, $\gamma_k$-covers) of $X$. An open cover of $X$ is called a $\gamma$-*cover* (respectively, a $\gamma_k$-*cover* [47] if each finite (respectively, compact) subset of $X$ belongs to all but finitely many elements of the cover.

$S_{fin}(\mathcal{O}, \mathcal{O})$ is the Menger property M.

$S_1(\mathcal{O}, \mathcal{O})$ is the Rothberger property R.

Recall the following definitions of classical star covering selection properties from [4] (see also [5,7]).

A space $X$ is said to be:

1. *Star Menger* SM (respectively, *star Rothberger* SR) if, for each sequence $(\mathcal{U}_n)_{n\in\mathbb{N}}$ of open covers of $X$, there is a sequence $(\mathcal{V}_n)_{n\in\mathbb{N}}$ (respectively, a sequence $(U_n)_{n\in\mathbb{N}}$) such that $\mathcal{V}_n$ is a finite subset of $\mathcal{U}_n$ (respectively, $U_n \in \mathcal{U}_n$) for each $n \in \mathbb{N}$, and $X = \bigcup_{n\in\mathbb{N}} \mathrm{St}(\bigcup \mathcal{V}_n, \mathcal{U}_n)$ (respectively, $X = \bigcup_{n\in\mathbb{N}} \mathrm{St}(U_n, \mathcal{U}_n)$).

2. *Strongly star Menger* SSM (respectively, *strongly star Rothberger* SSR) if, for each sequence $(\mathcal{U}_n)_{n\in\mathbb{N}}$ of open overs of $X$, there is a sequence $(F_n)_{n\in\mathbb{N}}$ of finite subsets of $X$ (respectively, a sequence $(x_n)_{n\in\mathbb{N}}$ of elements of $X$) such that $X = \bigcup_{n\in\mathbb{N}} \mathrm{St}(F_n, \mathcal{U}_n)$ (respectively, $X = \bigcup_{n\in\mathbb{N}} \mathrm{St}(x_n, \mathcal{U}_n)$).

3. *Star Hurewicz* SH (respectively, *strongly star Hurewicz* SSH) if, for each sequence $(\mathcal{U}_n)_{n\in\mathbb{N}}$ of open covers of $X$, there is a sequence $(\mathcal{V}_n)_{n\in\mathbb{N}}$ (respectively, a sequence $(F_n)_{n\in\mathbb{N}}$) such that $\mathcal{V}_n$ is a finite subset of $\mathcal{U}_n$ (respectively, $F_n$ is a finite subset of $X$) for each $n \in \mathbb{N}$, and each $x \in A$ belongs to all but finitely many sets $\mathrm{St}(\bigcup \mathcal{V}_n, \mathcal{U}_n)$ (respectively, to all but finitely many sets $\mathrm{St}(F_n, \mathcal{U}_n)$).

Quite recently, Caruvana and J. Holshouser [48] proposed a new approach to star selection principles which allows to consider star selection principles as standard classical selection principles. For this, they introduced two new operators Cons(tellation) (which is a collection of stars) and Gal(axy) (which is a collection of constellations).

In what follows, we will consider various generalizations and modifications of these properties. The paper is organized as follows. Section 2 is devoted to recent results on star selection principles in hyperspaces with various topologies. In Section 3, we discuss selective versions of covering properties (a) and acc, which can be viewed at the same time as generalizations of classical star selection principles. In Section 4, we present results on so-called set star selection properties. The paper ends by proposing lines of possible further study in the field of star selection principles. The list of references contains up-to-date works and gives coordinates about the papers, which can be useful to the interested reader.

## 2. Star Selection Principles and Hyperspaces

Let $X$ be a space. By $2^X$, we denote the family of closed subsets of $X$, and by $\mathsf{CL}(X)$, we denote the family of closed nonempty subsets of $X$. If $A$ is a subset of $X$ and $\mathcal{A}$ a family of subsets of $X$, then we use the notation

$$A^c = X \setminus A \text{ and } \mathcal{A}^c = \{A^c : A \in \mathcal{A}\},$$
$$A^- = \{F \in \mathsf{CL}(X) : F \cap A \neq \varnothing\}, \text{ and } \mathcal{A}^- = \{A^- : A \in \mathcal{A}\},$$
$$A^+ = \{F \in \mathsf{CL}(X) : F \subset A\} \text{ and } \mathcal{A}^+ = \{A^+ : A \in \mathcal{A}\}.$$

The most known and popular among topologies on $2^X$ and $\mathsf{CL}(X)$ is the Vietoris topology $\mathsf{V} = \mathsf{V}^- \vee \mathsf{V}^+$, where the *lower Vietoris topology* $\mathsf{V}^-$ is generated by all sets $A^-$, $A \subset X$ open, and the *upper Vietoris topology* $\mathsf{V}^+$ is generated by sets $B^+$, $B$ open in $X$ (see [49]).

Let $\Delta$ be a subset of $\mathsf{CL}(X)$. We consider only such $\Delta$ which is closed for finite unions and contains all singletons. Then the *upper $\Delta$-topology* [50], denoted by $\Delta^+$, is the topology whose subbase is the collection

$$\{(D^c)^+ : D \in \Delta\} \cup \{\mathsf{CL}(X)\}.$$

The $\Delta$-*topology* $\tau_\Delta$ has as a base the family

$$\left\{ \left( \bigcap_{i=1}^{m} V_i^- \right) \cap (B^c)^+ : B \in \Delta \text{ and } V_i \in \tau \text{ for } i \leq m \right\}.$$

Two important special cases are: 1. $\Delta$ is the family $\mathbb{F}(X)$ of all finite subsets of $X$, and 2. $\Delta$ is the collection $\mathbb{K}(X)$ of compact subsets of $X$. The corresponding $\Delta^+$-topologies ($\Delta$-topologies) will be denoted by $\mathsf{Z}^+$ ($\mathsf{Z}$) [51] and $\mathsf{F}^+$ ($\mathsf{F}$). The F-topology is known as the *Fell topology* (or the *co-compact topology*) [52], and the Z-topology is known as the co-finite topology.

The first study of selection principles in hyperspaces was carried out by P. Daniels for Pixley–Roy spaces PR($X$) over $X$ [53].

Recall that PR($X$) is the space $\mathbb{F}(X)$ with the topology whose base is the family

$$\{[A, U] : A \in \mathsf{PR}(X), U \text{ open in } X\},$$

where $[A, U] = \{B \in \mathsf{PR}(X) : A \subset B \subset U\}$.

A similar study was carried out in the papers [54–61].

In 2005, Kočinac started the study of selection properties of hyperspaces with the upper $\Delta$ topology, in particular, with the upper co-compact topology $\mathsf{F}^+$ and upper co-finite $\mathsf{Z}^+$ topologies. Di Maio, Kočinac and E. Meccariello [62,63] investigated $\mathsf{S}_1$ and $S_{fin}$-covering properties in $2^X$ and CL($X$) under $\mathsf{Z}^+$, $\mathsf{F}^+$, $\mathsf{V}^+$ and $\mathsf{V}^-$ topologies by using $k$-covers, $\omega$-covers and $\pi$-networks. In [64], some closure and convergence-selective properties were investigated.

Motivated by [62,63], Z. Li in his paper [65] introduced $k_F$-covers and $c_V$-covers (which are different from $k$-covers and $\omega$-covers) to establish selection principles in CL($X$) endowed with the Fell topology and the Vietoris topology. To explore selection principles in CL($X$) under $\tau_F$ and $\tau_V$, he introduced the following definitions of hit-and-miss type covers.

**Definition 1** ([65]). *An open cover $\mathcal{U}$ of $X$ ($X \notin \mathcal{U}$) is called a $k_F$-cover of $X$ if, for any compact subset $K$ of $X$ and any finite collection $V_1, V_2, \ldots, V_m$ of open sets in $X$ with $V_i \cap K^c \neq \varnothing$ ($1 \leq i \leq m$), there exist $U \in \mathcal{U}$ and a finite set $F \subset X$ with $F \cap V_i \neq \varnothing$ ($1 \leq i \leq m$) such that $K \subset U$ and $F \cap U = \varnothing$.*

*We denote by $\mathcal{K}_F(X)$ the collection of all $k_F$-covers of $X$.*

**Definition 2** ([65]). *Let $Y$ be a subset of $X$ with $Y \neq X$. An open cover $\mathcal{U}$ of $Y$ ($Y \notin \mathcal{U}$) is called a $k_F$-cover of $Y$ if, for any compact subset $K \subset Y$ and any finite family $V_1, V_2, \ldots, V_m$ of open sets in $X$ with $V_i \setminus Y \neq \varnothing$ ($1 \leq i \leq m$), there exist $U \in \mathcal{U}$ and a finite set $F$ with $F \cap V_i \neq \varnothing$ ($1 \leq i \leq m$) such that $K \subset U$ and $F \cap U = \varnothing$.*

*The collection of all $k_F$-covers of $Y$ with $Y \neq X$ is denoted by $\mathcal{K}_F(Y, X)$.*

**Definition 3** ([65]). *An open cover $\mathcal{U}$ of $X$ ($X \notin \mathcal{U}$) is called a $c_V$-cover of $X$ if, for any finite family $V_1, V_2, \ldots, V_m$ of open sets in $X$, there exist $U \in \mathcal{U}$ and a finite set $F$ with $F \cap V_i \neq \varnothing$ ($1 \leq i \leq m$) such that $\bigcap_{i=1}^{m} V_i^c \subset U$ and $F \cap U = \varnothing$.*

*The collection of all $c_V$-covers of $X$ is denoted by $\mathcal{C}_V(X)$.*

**Definition 4** ([65]). *Let $Y$ be a subset of $X$ with $Y \neq X$. An open cover $\mathcal{U}$ of $Y$ ($Y \notin \mathcal{U}$) is called a $c_V$-cover of $Y$ if, for any family $V_1, V_2, \ldots, V_m$ of open sets in $X$ with $\bigcap_{i=1}^{m} V_i^c \subset Y$ and $V_i \setminus Y \neq \varnothing$ ($1 \leq i \leq m$), there exist $U \in \mathcal{U}$ and a finite set $F$ with $F \cap V_i \neq \varnothing$ ($1 \leq i \leq m$) such that $\bigcap_{i=1}^{m} V_i^c \subset U$ and $F \cap U = \varnothing$.*

*The collection of all $c_V$-covers of $Y$ with $Y \neq X$ is denoted by $\mathcal{C}_V(Y, X)$.*

Li also defined $\pi_F$-networks and $\pi_V$-networks of a space $X$.

**Definition 5** ([65]). *(1) A family* $\xi = \{(K; V_1, V_2, \ldots V_m) : K$ *a compact subset of* $X, V_1, V_2, \ldots V_m$ *open subsets of* $X$ *with* $V_i \cap K^c \neq \varnothing, i \leq m, m \in \mathbb{N}\}$ *is called a* $\pi_F$-network *of* $X$ *if, for each open set* $U \subset X$ *with* $U \neq X$, *there exist a* $(K; V_1, V_2, \ldots V_m) \in \xi$ *and a finite set* $F \subset X$ *with* $F \cap V_i \neq \varnothing, i \leq m$, *such that* $K \subset U$ *and* $F \cap U = \varnothing$.

$\Pi_F$ *denotes the collection of* $\pi_F$ *networks of* $X$.

*(2) A family* $\zeta = \{(V_1, V_2, \ldots V_k) : V_1, V_2, \ldots V_m$ *open subsets of* $X, m \in \mathbb{N}\}$ *is called a* $\pi_V$-network *of* $X$ *if, for each open set* $U \subset X$ *with* $U \neq X$, *there exist a* $(V_1, V_2, \ldots, V_m) \in \zeta$ *and a finite set* $F \subset X$ *with* $F \cap V_i \neq \varnothing, i \leq m$, *such that* $\bigcap_{i \leq m} V_i^c \subset U$ *and* $F \cap U = \varnothing$.

$\Pi_V$ *denotes the collection of* $\pi_V$-networks *of* $X$.

By using $k_F$-covers, $c_V$-covers, $\pi_F$-networks, and $\pi_V$-networks, he investigated and characterized the Menger, Rothberger (and Hurewicz and some other properties) properties of $\mathsf{CL}(\mathsf{X})$ with the topologies $\mathsf{V}$ and $\mathsf{F}$.

The notions of $k_F$-covers, $c_V$-covers, $\pi_F$-networks, and $\pi_V$-networks have been generalized in [29,66,67] in terms of subsets of $\mathsf{CL}(X)$.

In [67], the authors generalized $k_F$-covers and $c_V$-covers and defined the covers that they called $\Delta_F$-covers, $\Delta \subset \mathsf{CL}(X)$, which reduce to $k_F$-covers and $c_V$-covers for special $\Delta$. The notion of $\Delta_F$-covers is obtained by replacing in Definition 1 "*K* compact" with "$K \in \Delta$". By using this generalization, two important properties in selection principles theory, the Reznichenko and Pytkeev properties, have been characterized in hyperspaces equipped with the $\tau_\Delta$ topology. We do not present these characterizations because they are not related to the theory of star selection principles.

In [66], the authors introduced the notion of a $\pi_F(\Delta)$-network, $\Delta \subset \mathsf{CL}(X)$, by replacing in Definition 5(1) of a $\pi_F$-network "*U* is open" by "$U \in \Delta^c$", and using it, they characterized the SR and SSR properties in hyperspaces endowed with the Fell topology. For this, they introduced two selection principles, as follows.

**Definition 6.** *(1)* $\mathsf{FELL}(\Pi_F(\Delta), \Pi_F(\Delta))$: *For each sequence* $(J_n : n \in \mathbb{N})$ *of elements of* $\Pi_F(\Delta)$, *there is a sequence* $(U_n : n \in \mathbb{N})$ *of elements of* $\Delta^c$ *such that* $\mathcal{J} = \bigcup_{n \in \mathbb{N}} \{(K_s^n; V_{1,s}^n, \ldots, V_{m_s,s}^n) \in J_n : K_s^n \subset U_n, V_{i,s}^n \nsubseteq U_n, i \leq m_s\}$ *belongs to* $\Pi_F(\Delta)$.

*(2)* $\mathsf{FELL}^*(\Pi_F(\Delta), \Pi_F(\Delta))$: *For each sequence* $(J_n = \{(K_s^n; V_{1,s}^n, \ldots, V_{m_s,s}^n) : s \in S_n\} : n \in \mathbb{N})$ *of elements of* $\Pi_F(\Delta)$, *for each* $n$ *there is* $s_n \in S_n$ *such that* $\mathcal{J} = \bigcup_{n \in \mathbb{N}} \{(K_s^n; V_{1,s}^n, \ldots, V_{m_s,s}^n) :$ *there is* $U \in \Delta^c$, *such that* $(K_s^n \cup K_{s_n}^n) \subset U, V_{i,s}^n \nsubseteq U, i \leq m_s$ *and* $V_{j,s_n}^n \nsubseteq U, j \leq m_{s_n}\}$ *belongs to* $\Pi_F(\Delta)$.

Then, they proved the following two theorems.

**Theorem 1** ([66], Theorem 2.4). *The following are equivalent:*

(1) *The hyperspace* $(\Delta, \mathsf{F})$ *is* SR;
(2) *X satisfies the selection principle* $\mathsf{FELL}^*(\Pi_F(\Delta), \Pi_F(\Delta))$.

**Theorem 2** ([66], Theorem 2.2). *The following are equivalent:*

(1) *The hyperspace* $(\Delta, \mathsf{F})$ *is* SSR;
(2) *X satisfies the selection principle* $\mathsf{FELL}(\Pi_F(\Delta), \Pi_F(\Delta))$.

Important consequences of these two results are the following two corollaries.

**Corollary 1.** *For a space X, the following hold:*

(1) $(\mathsf{CL}(X), \mathsf{F})$ *is* SR *if and only if X satisfies* $\mathsf{FELL}^*(\Pi_F, \Pi_F)$;
(2) $(\mathbb{K}(X), \mathsf{F})$ *is* SR *if and only if X satisfies* $\mathsf{FELL}^*(\Pi_F(\mathbb{K}(X), \Pi_F(\mathbb{K}(X))))$;
(3) $(\mathbb{F}(X), \mathsf{F})$ *is* SR *if and only if X satisfies* $\mathsf{FELL}^*(\Pi_F(\mathbb{F}(X), \Pi_F(\mathbb{F}(X))))$.

**Corollary 2.** *For a space X, the following hold:*

(1) $(\mathsf{CL}(X), \mathsf{F})$ *is* SSR *if and only if X satisfies* $\mathsf{FELL}(\Pi_F, \Pi_F)$;
(2) $(\mathbb{K}(X), \mathsf{F})$ *is* SSR *if and only if X satisfies* $\mathsf{FELL}(\Pi_F(\mathbb{K}(X), \Pi_F(\mathbb{K}(X))))$;
(3) $(\mathbb{F}(X), \mathsf{F})$ *is* SSR *if and only if X satisfies* $\mathsf{FELL}(\Pi_F(\mathbb{F}(X), \Pi_F(\mathbb{F}(X))))$.

A similar scenario was applied in [68] for characterizations of the properties SM and SSM. For this, the authors first introduced the following selection principles.

**Definition 7.** *(1)* $\mathsf{FELL}_M(\Pi_F(\Delta), \Pi_F(\Delta))$: *For each sequence* $(J_n : n \in \mathbb{N})$ *of elements of* $\Pi_F(\Delta)$, *there is a sequence* $(\mathcal{U}_n : n \in \mathbb{N})$ *of finite subsets of* $\Delta^c$ *with* $\mathcal{U}_n \neq \varnothing$, $n \in \mathbb{N}$, *such that* $\mathcal{J} = \bigcup_{n \in \mathbb{N}} \{(K_s^n; V_{1,s}^n, \dots, V_{m_s,s}^n) \in J_n : \text{ there is } U \in \mathcal{U}_n \text{ such that } K_s^n \subset U, V_{i,s}^n \nsubseteq U, i \leq m_s\}$ *belongs to* $\Pi_F(\Delta)$.

*(2)* $\mathsf{FELL}_M^*(\Pi_F(\Delta), \Pi_F(\Delta))$: *For each sequence* $(J_n = \{(K_s^n; V_{1,s}^n, \dots, V_{m_s,s}^n) : s \in S_n\} : n \in \mathbb{N})$ *of elements of* $\Pi_F(\Delta)$ *for each n there are finite* $T_n \subset S_n$, $n \in \mathbb{N}$, *such that* $\mathcal{J} = \bigcup_{n \in \mathbb{N}} \{(K_s^n; V_{1,s}^n, \dots, V_{m_s,s}^n) \in J_n : \text{ there are } U \in \Delta^c \text{ and } s_n \in T_n \text{ such that } (K_s^n \cup K_{s_n}^n) \subset U, V_{i,s}^n \nsubseteq U, i \leq m_s \text{ and } V_{j,s_n}^n \nsubseteq U, j \leq m_{s_n}\}$ *belongs to* $\Pi_F(\Delta)$.

The characterizations of SM and SSM in hyperspaces with the Fell topology are given in the next two theorems.

**Theorem 3** ([68]). *The following are equivalent:*

(1) *The hyperspace* $(\Delta, \mathsf{F})$ *is* SM;
(2) *X satisfies the selection principle* $\mathsf{FELL}_M^*(\Pi_F(\Delta), \Pi_F(\Delta))$.

**Theorem 4** ([68], Theorem 2.2). *The following are equivalent:*

(1) *The hyperspace* $(\Delta, \mathsf{F})$ *is* SSM;
(2) *X satisfies the selection principle* $\mathsf{FELL}_M(\Pi_F(\Delta), \Pi_F(\Delta))$.

Corollaries of these two results are:

**Corollary 3.** *For a space X, the following hold:*

(1) $(\mathsf{CL}(X), \mathsf{F})$ *is* SM *if and only if X satisfies* $\mathsf{FELL}_M^*(\Pi_F, \Pi_F)$;
(2) $(\mathbb{K}(X), \mathsf{F})$ *is* SM *if and only if X satisfies* $\mathsf{FELL}_M^*(\Pi_F(\mathbb{K}(X), \Pi_F(\mathbb{K}(X))))$;
(3) $(\mathbb{F}(X), \mathsf{F})$ *is* SM *if and only if X satisfies* $\mathsf{FELL}_M^*(\Pi_F(\mathbb{F}(X), \Pi_F(\mathbb{F}(X))))$.

**Corollary 4.** *For a space X, the following hold:*

(1) $(\mathsf{CL}(X), \mathsf{F})$ *is* SSM *if and only if X satisfies* $\mathsf{FELL}_M(\Pi_F, \Pi_F)$;
(2) $(\mathbb{K}(X), \mathsf{F})$ *is* SSM *if and only if X satisfies* $\mathsf{FELL}_M(\Pi_F(\mathbb{K}(X), \Pi_F(\mathbb{K}(X))))$;
(3) $(\mathbb{F}(X), \mathsf{F})$ *is* SSM *if and only if X satisfies* $\mathsf{FELL}_M(\Pi_F(\mathbb{F}(X), \Pi_F(\mathbb{F}(X))))$.

The following is an interesting result on the coincidence of SSM and SSR properties on hyperspaces.

**Theorem 5** ([68]). *Let X be a space and* $\Delta \subset \mathsf{CL}(X)$. *The following are equivalent:*

(1) $(\Delta, \mathsf{V}^-)$ *is* SSM;
(2) $(\Delta, \mathsf{V}^-)$ *is* SSR.

In [4], the following general form of a star selection principle was introduced. Let $\mathcal{A}$ and $\mathcal{B}$ be collections of some kind of open covers of space $X$ and $\mathcal{M}$ be a family of subsets of $X$. Then the symbol $SS_\mathcal{M}^*(\mathcal{A}, \mathcal{B})$ denotes the selection principle that for each sequence $(\mathcal{U}_n : n \in \mathbb{N})$ of elements of $\mathcal{A}$, there is a sequence $(M_n : n \in \mathbb{N})$ of elements of $\mathcal{M}$ such that $\{\mathsf{St}(M_n, \mathcal{U}_n) : n \in \mathbb{N}\} \in \mathcal{B}$. If $\mathcal{M}$ is the collection of compact subsets of $X$, then the spaces satisfying $SS_\mathcal{M}^*(\mathcal{O}, \mathcal{O})$ are called *star-K-Menger* in [69].

In ([66], Theorem 3.4), the following was proved.

**Theorem 6.** *For a space $X$ and $\Delta \subset \mathsf{CL}(X)$, the following statements are equivalent:*

(1) *$(\Delta, \mathsf{V}^-)$ is SSR;*

(2) *$X$ satisfies $\mathsf{SS}^*_\Delta(\mathcal{O}, \mathcal{O})$.*

For $\Delta \in \{\mathbb{K}(X), \mathbb{F}(X)\}$, the following corollaries of Theorem 6 are obtained.

**Corollary 5.** *For a space $X$, the following are true:*

(1) *$(\mathbb{K}(X), \mathsf{V}^-)$ is SSR if and only if $X$ is star-K-Menger;*

(2) *$(\mathbb{F}(X), \mathsf{V}^-)$ is SSR if and only if $X$ is SSM.*

From here, together with Theorem 5 and its corollaries, one obtains the following results.

**Corollary 6.** *For a space $X$, the following hold:*

(1) *$(\mathbb{K}(X), \mathsf{V}^-)$ is SSM;*

(1) *$(\mathbb{K}(X), \mathsf{V}^-)$ is SSR;*

(3) *$X$ is star-K-Menger.*

**Corollary 7.** *For a space $X$, the following hold:*

(1) *$(\mathbb{F}(X), \mathsf{V}^-)$ is SSM;*

(1) *$(\mathbb{F}(X), \mathsf{V}^-)$ is SSR;*

(3) *$X$ is SSM.*

In [70], the authors generalized the notion of a $\pi_V$-network and introduced the notion of a $\pi_V(\Delta)$-network as follows. Let $\Delta \subset \mathsf{CL}(X)$ and $\xi = \{(U_1, U_2, \ldots, U_m) : U_i \text{ open}, m \in \mathbb{N}\}$. Then, $\xi$ is said to be a $\pi_V(\Delta)$-*network* for $X$ if, for each $U \in \Delta^c$, there exist a $(V_1, \ldots, V_m) \in \xi$ and a finite $F \subset X$ intersecting each $V_i$, $i \leq m$, such that $\bigcap_{i=1}^m V_i^c \subset U$ and $F \cap U = \emptyset$.

$\Pi_V(\Delta)$ denotes the collection of $\pi_V(\Delta)$-networks of $X$.

The authors also introduced two technical selection principles.

**Definition 8.** *(1) $\mathsf{S}_{\Pi_V}(\Pi_V(\Delta), \Pi_V(\Delta))$: For each sequence $(J_n : n \in \mathbb{N})$ of elements of $\Pi_V(\Delta)$, there is a sequence $(U_n : n \in \mathbb{N})$ of elements of $\Delta^c$ such that $\mathcal{J} = \bigcup_{n \in \mathbb{N}} \{(V_{1,s}^n, \ldots, V_{m_s,s}^n) \in J_n : \bigcap_{i \leq m_s} (V_{i,s}^n)^c \subset U_n, V_{i,s}^n \nsubseteq U_n, i \leq m_s\}$ belongs to $\Pi_V(\Delta)$.*

*(2) $\mathsf{S}^*_{\Pi_V}(\Pi_V(\Delta), \Pi_V(\Delta))$: For each sequence $(J_n = \{(V_{1,s}^n, \ldots, V_{m_s,s}^n) : s \in S_n\} : n \in \mathbb{N})$ of elements of $\Pi_V(\Delta)$, for each $n$, there are $s_n \in S_n$, $n \in \mathbb{N}$, such that $\mathcal{J} = \bigcup_{n \in \mathbb{N}} \{(V_{1,s}^n, \ldots, V_{m_s,s}^n) \in J_n : \text{there is } U \in \Delta^c \text{ such that } (\bigcap_{i \leq m_s} (V_{i,s}^n)^c) \cup (\bigcap_{i \leq m_{s_n}} (V_{i,s_n}^n)^c) \subset U, V_{i,s}^n \nsubseteq U, i \leq m_s, V_{j,s_n}^n \nsubseteq U, j \leq m_{s_n}\}$ belongs to $\Pi_F(\Delta)$.*

**Theorem 7** ([70], Theorem 2.11). *Given a topological space $X$, the following conditions are equivalent:*

(1) *$(\Delta, \mathsf{V})$ is SR;*

(2) *$X$ satisfies $\mathsf{S}^*_{\Pi_V}(\Pi_V(\Delta), \Pi_V(\Delta))$.*

**Corollary 8** ([70], Corollary 2.12). *Let $X$ be a topological space, and let $\Delta$ be one of the following hyperspaces: $\mathsf{CL}(X), \mathbb{K}(X), \mathbb{F}(X)$. Then, $(\Delta, \mathsf{V})$ is SR if and only if $X$ satisfies $\mathsf{S}_{\Pi_V}(\Pi_V(\Delta), \Pi_V(\Delta))$.*

**Theorem 8** ([70], Theorem 2.8). *Given a topological space $X$, the following conditions are equivalent:*

(1) *$(\Delta, \mathsf{V})$ is SSR;*

(2) *$X$ satisfies $\mathsf{S}_{\Pi_V}(\Pi_V(\Delta), \Pi_V(\Delta))$.*

**Corollary 9** ([70], Corollary 2.9). *Let X be a topological space, and let $\Delta$ be one of the following hyperspaces: CL(X), $\mathbb{K}(X)$, $\mathbb{F}(X)$ or $\mathbb{CS}(X)$ (the family of all convergent sequences in X). Then, $(\Delta, \mathsf{V})$ is SSR if and only if X satisfies $\mathsf{S}_{\Pi_V}(\Pi_V(\Delta), \Pi_V(\Delta))$.*

Further generalizations have been made in the papers [29,71,72], where the authors consider two subsets $\Gamma$ and $\Delta$ of CL(X) and define $c_\Delta(\Gamma)$-covers and $\pi_\Delta(\Gamma)$-networks to investigate the hyperspaces $(\Lambda, \tau_\Delta)$. As in the results above, they introduce a few technical selection principles for a space X and then obtain a number of interesting results on the star selection principles of Rothberger- and Menger-type (and other properties) in hyperspaces in terms of the mentioned selection principles. Some of these results generalize the results presented above, while some are completely new.

At the end of this section, we notice that the papers [73,74] contain a series of nice results on selective two-person infinitely long games in hyperspaces, naturally corresponding to selection principles.

Additionally, we would like to mention that the paper [75] gives interesting connections between classical selection principles and star selection principles.

## 3. Selective Version of the acc and (a) Properties

The following subclass of the class of countably compact spaces was defined and studied by Matveev in [76]. A space X is said to be an *absolutely countably compact space* (shortly acc-*space*) if, for each open cover $\mathcal{U}$ of X and each dense subset D of X, there exists a finite set $K \subset D$ such that $\mathrm{St}(K, \mathcal{U}) = X$. Matveev [77] also introduced a property which is a a generalization of countable compactness: a space X is said to be an (a)-*space* if, for any open cover $\mathcal{U}$ of X and any dense subset D of X, there is a closed discrete (in X) set $K \subset D$ such that $\mathrm{St}(K, \mathcal{U}) = X$.

There are three natural ways to define selective versions of these properties.

1. For each sequence $(\mathcal{U}_n : n \in \mathbb{N})$ of open covers of X and each dense subset D of X, there is a sequence $(K_n : n \in \mathbb{N})$ of finite (resp., closed discrete) subsets of D such that $\bigcup_{n \in \mathbb{N}} \mathrm{St}(K_n, \mathcal{U}_n) = X$.

2. For each sequence $(\mathcal{U}_n : n \in \mathbb{N})$ of open covers of X and each sequence $(D_n : n \in \mathbb{N})$ of dense subsets of X, there is a sequence $(K_n : n \in \mathbb{N})$ of finite (resp. closed discrete) subsets of $D_n$, $n \in \mathbb{N}$, such that $\bigcup_{n \in \mathbb{N}} \mathrm{St}(K_n, \mathcal{U}_n) = X$.

3. For each open cover $\mathcal{U}$ of X and each sequence $(D_n : n \in \mathbb{N})$ of dense subsets of X, there is a sequence $(K_n : n \in \mathbb{N})$ of finite (resp., closed discrete) subsets of $D_n$, $n \in \mathbb{N}$ such that $\bigcup_{n \in \mathbb{N}} \mathrm{St}(K_n, \mathcal{U}_n) = X$.

The first approach was applied in [78], and the second one was applied in several papers that appeared recently (see [79–85]). The third approach was applied in [86] under the name selectively star-Lindelöf spaces: a space X is *selectively star-Lindelöf* if, for any open cover $\mathcal{U}$ of X and any sequence $(D_n : n \in \mathbb{N})$ of dense subsets of X, there are finite sets $F_n \subset D_n$, $n \in \mathbb{N}$, such that $\mathrm{St}(\bigcup_{n \in \mathbb{N}} F_n, \mathcal{U}) = X$. Then, these spaces have been studied in more detail in [79,87] (where the authors use the name *selectively absolutely star-Lindelöf spaces*). Quite recently, in [83], the authors studied Hurewicz-type spaces that they call H-star-Lindelöf.

In this paper, we define selective versions of the properties (a) and acc following a general idea in the star selection principles theory [4,7].

First, we give the following general selective version of the notions of acc-spaces and (a)-spaces following the terminology and notation in [84,85].

**Definition 9.** *Let X be a space. Denote by $\mathcal{A}$ and $\mathcal{B}$ collections of some open covers of X, and by $\mathcal{C}$ a collection of subsets of X. Then, X is said to be a* strictly selectively $(\mathcal{A}, \mathcal{B})$-$(a)_{\mathcal{C}}$-space, *denoted by $X \in \mathsf{StrSel}(\mathcal{A}, \mathcal{B})$-$(a)_{\mathcal{C}}$, if, for each sequence $(\mathcal{U}_n : n \in \mathbb{N})$ of elements of $\mathcal{A}$ and each sequence $(D_n : n \in \mathbb{N})$ of dense subsets of X, there is a sequence $(K_n : n \in \mathbb{N})$ of elements of $\mathcal{C}$ such that each $K_n$ is a subset of $D_n$ and $\{\mathrm{St}(K_n, \mathcal{U}_n) : n \in \mathbb{N}\} \in \mathcal{B}$.*

In this way, we obtain several classes of spaces. The spaces satisfying

(1) $\text{StrSel}(\mathcal{O}, \mathcal{O})\text{-}(a)_{\text{finite}}$ we call *Menger* acc-*spaces* (shortly, M-acc-spaces);

(2) $\text{StrSel}(\mathcal{O}, \Gamma)\text{-}(a)_{\text{finite}}$ we call *Hurewicz* acc-*spaces* (shortly H-acc-spaces);

(3) $\text{StrSel}(\mathcal{O}, \mathcal{O})\text{-}(a)_{\text{singleton}}$ we call *Rothberger* acc-*spaces* (shortly, R-acc-spaces);

(4) $\text{StrSel}(\mathcal{O}, \Omega)\text{-}(a)_{\text{finite}}$ are called $\omega$-*Menger* acc-*spaces* (shortly, $\omega$-M-acc-spaces);

(5) $\text{StrSel}(\mathcal{O}, \Omega)\text{-}(a)_{\text{singleton}}$ are called $\omega$-*Rothberger* acc-*spaces* (shortly, $\omega$-R-acc-spaces);

(6) $\text{StrSel}(\mathcal{O}, \Gamma_k)\text{-}(a)_{\text{finite}}$ are called *k-Hurewicz* acc-*spaces* (shortly, k-H-acc-spaces);

(7) $\text{StrSel}(\mathcal{O}, \mathcal{O})\text{-}(a)_{\text{closeddiscrete}}$ will be called *strictly selectively* (*a*)-*spaces*.

Observe that M-acc spaces have already been studied in [79] using the name *selectively strongly star-Menger spaces*. The relations of this class of spaces with absolutely strongly star Menger spaces (shortly, aSSM-spaces [78] and the class of selectively absolutely star-Lindelöf spaces [3] are given in the following diagram:

$$\text{M} \Rightarrow \text{M-acc} \Rightarrow \text{aSSM} \Rightarrow \text{SSM} \Rightarrow \text{SM}.$$

Notice that in the class of Hausdorff paracompact spaces, all these classes are equivalent.

Additionally, let us mention that recently (and independently), the classes of H-acc and R-acc spaces have been considered under different names in [82].

In particular, the paper [82] contains interesting results on cardinality restrictions of the form $|X|$, being at most small combinatorial cardinals $\mathfrak{d}$, $\mathfrak{b}$, and $\mathfrak{cov}(\mathcal{M})$.

For the convenience of the reader, we give definitions of small combinatorial cardinals. Let $\mathbb{N}^{\mathbb{N}}$ be the countable Tychonoff power of the discrete space $D(\omega)$. A natural pre-order $\prec^*$ on $\mathbb{N}^{\mathbb{N}}$ is defined by $f \prec^* g$ if and only if $f(n) \leq g(n)$ for all but finitely many $n$. A subset $F$ of $\mathbb{N}^{\mathbb{N}}$ is said to be *dominating* if, for each $g \in \mathbb{N}^{\mathbb{N}}$, there is a function $f \in F$ such that $g \prec^* f$. A subset $F$ of $\mathbb{N}^{\mathbb{N}}$ is called *bounded* if there is an $g \in \mathbb{N}^{\mathbb{N}}$ such that $f \prec^* g$ for each $f \in F$. The symbol $\mathfrak{b}$ (resp. $\mathfrak{d}$) denotes the least cardinality of an unbounded (resp. dominating) subset of $\mathbb{N}^{\mathbb{N}}$. Another uncountable small cardinal is the cardinal $\mathfrak{cov}(\mathcal{M})$, the *covering number of the ideal of meager subsets of* $\mathbb{R}$ characterized in terms of subsets of $\mathbb{N}^{\mathbb{N}}$

$$\mathfrak{cov}(\mathcal{M}) = \min\{|F| : F \subset \mathbb{N}^{\mathbb{N}} \text{ such that } \forall g \in \mathbb{N}^{\mathbb{N}} \exists f \in F \text{ with } f(n) \neq g(n) \forall n \in \mathbb{N}\}.$$

We also need the following: a space $X$ is said to be *absolutely strongly star Lindelöf* aSSL if, for each open cover $\mathcal{U}$ of $X$ and each dense subset $D$ of $X$, there is a countable set $C \subset D$ such that $\text{St}(C, \mathcal{U}) = X$ [3].

**Theorem 9** ([82]). *The following assertions hold:*

(1) *If a space $X$ of cardinality less than $\mathfrak{d}$ is* aSSL, *then it is* M-acc;

(2) *If a space $X$ of cardinality less than $\mathfrak{b}$ is* aSSL, *then it is* H-acc;

(3) *If a space $X$ of cardinality less than $\mathfrak{cov}(\mathcal{M})$ is* aSSL, *then it is* R-acc-*space.*

Recall that $e(X)$ is the *extent* of a space $X$, the supremum of cardinalities of closed discrete subsets of $X$.

**Theorem 10** ([82]). *Let $X$ be a space of countable extent. Then, the following hold:*

(1) *If $|X| < \mathfrak{d}$, then $X$ is selectively (a) if and only if $X$ is* M-acc;

(2) *If $|X| < \mathfrak{b}$, then $X$ is selectively (a) if and only if $X$ is* H-acc;

(3) *If $|X| < \mathfrak{cov}(\mathcal{M})$, then $X$ is selectively (a) if and only if $X$* R-acc.

We are going now to present a few properties of the classes M-acc and R-acc.

The following topological construction is well-known. Let $(X, \tau)$ be a topological space. The *Alexandroff duplicate* of $X$ (see [1]) is the set $\text{AD}(X) := X \times \{0, 1\}$ equipped with the following topology:

(i) All points $(x, 1)$, $x \in X$, are isolated;

(ii) Points $(x, 0)$ have a local base of the form $(U \times \{0, 1\}) \setminus \{(x, 1)\}$, where $U$ is open in $X$ and $x \in U$.

For many topological properties $\mathcal{P}$, the space AD($X$) has $\mathcal{P}$ if $X$ has $\mathcal{P}$. Such covering properties are, for example, compactness, Lindelöfness, and (hereditary) paracompactness.

We investigate similar questions for the classes defined above.

In the sequel, we denote by $I_X$ the set of isolated points of a space $X$.

The following fact is used in the sequel.

**Lemma 1.** *Each dense subset of* AD($X$) *contains the set* $(I_X \times \{0\}) \cup (X \times \{1\})$.

**Theorem 11** ([84,85]). *If the product $X \times Y$ of a space $X$ and a compact space $Y$ is strictly selectively $(a)$, then $X$ is strictly selectively $(\mathcal{O}, \mathcal{O})$-$(a)_{\text{closed}}$.*

**Problem 1.** *Is the product of a strictly selectively $(a)$-space $X$ and a metrizable compact space $Y$ strictly selectively $(a)$?*

**Theorem 12** ([84,85]). *If $X \in \mathsf{StrSel}(\mathcal{O}, \mathcal{O})$-$(a)_{\text{discrete}}$ and $e(\mathsf{AD}(X)) < \omega_1$, then $\mathsf{AD}(X)$ is also in $\mathsf{StrSel}(\mathcal{O}, \mathcal{O})$-$(a)_{\text{discrete}}$.*

Notice that a result similar to Theorem 12 was obtained in ([78], Theorem 2.9).

**Theorem 13** ([84,85]). *If a space $X$ is M-acc and $e(\mathsf{AD}(X)) < \omega_1$, then $\mathsf{AD}(X)$ is also M-acc.*

For $p \in \mathbb{N}$, denote by $\mathsf{M_p-acc}$ the class of M-acc spaces such that finite sets $A_n \subset D_n$ in the definition of M-acc spaces have, at most, $p$ elements.

The reader can find the proof of the following theorem in [85].

**Theorem 14** ([84,85]). *If a space $X$ is R-acc and $e(\mathsf{AD}(X)) < \omega_1$, then $\mathsf{AD}(X)$ is $\mathsf{M_3-acc}$.*

The following two theorems have been announced without proofs in [85].

**Theorem 15.** *If the Alexandroff duplicate* AD($X$) *of a space $X$ is strictly selectively $(\mathcal{O}, \mathcal{O}) - (a)_{\text{countable}}$, then $e(X) < \omega_1$.*

**Proof.** Suppose to the contrary that there is a closed discrete subset $B$ of $X$ having cardinality $\geq \omega_1$. The set $B \times \{1\}$ is closed and open in AD($X$). For each $n \in \mathbb{N}$ let $A_n = (B \times \{1\}) \setminus (C_n \times \{1\})$, where each $C_n$ is a countable subset of $B$. Every $A_n$ is a closed (discrete) subset of AD($X$). For each $n$, define $\mathcal{U}_n = (\mathsf{AD}(X) \setminus A_n) \times \{\{(x,1)\} : (x,1) \in A_n\}$. We claim that the sequence $(\mathcal{U}_n : n \in \mathbb{N})$ of open covers of AD($X$) and the dense set $D = (I_X \times \{0\}) \cup (X \times \{1\}) \subset \mathsf{AD}(X)$ witness that AD($X$) is not strictly selectively $(\mathcal{O}, \mathcal{O})$-$(a)_{\text{countable}}$. Indeed, if $(F_n : n \in \mathbb{N})$ is a sequence of countable sets with $F_n \subset D_n$, then there is a point $b \in B$ such that $(b, 1) \notin \bigcup_{n \in \mathbb{N}} F_n$. Since $(b, 1)$ is an isolated point in AD($X$), the set $\{(b, 1)\}$ is the only element of every $\mathcal{U}_n$ that contains $(b, 1)$, and $(b, 1) \notin \mathsf{St}(F_n, \mathcal{U}_n)$ for each $n \in \mathbb{N}$. This contradicts the assumption on AD($X$). $\square$

**Theorem 16.** *Let $A$ and $B$ be subspaces of a space $X$ such that $\overline{A} \cap B = \emptyset$ and $Z = (A \times \{1\}) \cup (B \times \{0\})$. If $e(Z) < \omega_1$ and $B$ is strictly selectively $(\mathcal{O}, \mathcal{O})$-$(a)_{\text{discrete}}$, then $Z$ is strictly selectively $(\mathcal{O}, \mathcal{O})$-$(a)_{\text{discrete}}$.*

**Proof.** Let $(\mathcal{U}_n : n \in \mathbb{N})$ be a sequence of open covers of $Z$ and let $(D_n : n \in \mathbb{N})$ be a sequence of dense subsets of $Z$. One may suppose that $D_n = (E_n \times \{0\}) \cup (A \times \{1\})$, where $E_n$ is a dense subset of $B \setminus A$ because every dense subset of $Z$ contains the set $D_n$. Thus, $E_n$ is dense in $B$. It remains to repeat the proof of Theorem 12 (or of the proof of Theorem 2.9 in [78]) with small appropriate changes. $\square$

### 4. Set Star Selection Properties

Let $X$ be a topological space, $A$ be a subset of $X$, and $\mathcal{C}$ be a collection of subsets of $X$. It is understood that properties of $A$ or of members of the collection $\mathcal{C}$ depend on their location in $X$. Thus, we can speak about relative properties of $A$ or of elements of $\mathcal{C}$ in $X$. In other words, we have the following:

A.V. Arhangel'skii initiated this kind of investigation in [88,89].

A similar line of investigation was recently proposed in the theory of (star) selection principles (see, [90–94]).

The set star selection properties (in a very general form) were first presented in .

**Definition 10** ([93]). *Let $\mathcal{C}$ be a family of nonempty subsets of a space $X$. We say that $X$ is:*

(1) *$\mathcal{C}$-star Menger (respectively, weakly $\mathcal{C}$-star Menger, almost $\mathcal{C}$-star Menger, faintly $\mathcal{C}$-star Menger) if for each $A \in \mathcal{C}$ and each sequence $(\mathcal{U}_n)_{n\in\mathbb{N}}$ of covers of $\overline{A}$ by sets open in $X$, there is a sequence $(\mathcal{V}_n)_{n\in\mathbb{N}}$ such that $\mathcal{V}_n$ is a finite subset of $\mathcal{U}_n$ for each $n \in \mathbb{N}$, and $A \subset \bigcup_{n\in\mathbb{N}}\mathsf{St}(\bigcup\mathcal{V}_n,\mathcal{U}_n)$ (respectively, $A \subset \overline{\bigcup_{n\in\mathbb{N}}\mathsf{St}(\bigcup\mathcal{V}_n,\mathcal{U}_n)}$, $A \subset \bigcup_{n\in\mathbb{N}}\mathsf{St}(\bigcup\mathcal{V}_n,\mathcal{U}_n)$, $A \subset \bigcup_{n\in\mathbb{N}}\mathsf{St}(\overline{\bigcup\mathcal{V}_n},\mathcal{U}_n))$;*

(2) *$\mathcal{C}$-strongly star Menger (respectively, weakly $\mathcal{C}$-strongly star Menger, almost $\mathcal{C}$-strongly star Menger) if, for each $A \in \mathcal{C}$ and each sequence $(\mathcal{U}_n)_{n\in\mathbb{N}}$ of covers of $\overline{A}$ by sets open in $X$, there is a sequence $(F_n)_{n\in\mathbb{N}}$ of finite subsets of $\overline{A}$ such that $A \subset \bigcup_{n\in\mathbb{N}}\mathsf{St}(F_n,\mathcal{U}_n)$ (respectively, $A \subset \overline{\bigcup_{n\in\mathbb{N}}\mathsf{St}(F_n,\mathcal{U}_n)}$, $A \subset \bigcup_{n\in\mathbb{N}}\mathsf{St}(F_n,\mathcal{U}_n)$).*

(3) *$\mathcal{C}$-star Rothberger (respectively, weakly $\mathcal{C}$-star Rothberger, almost $\mathcal{C}$-star Rothberger, faintly $\mathcal{C}$-star Rothberger) if, for each $A \in \mathcal{C}$ and each sequence $(\mathcal{U}_n)_{n\in\mathbb{N}}$ of collections of sets open in $X$ such that $\overline{A} \subset \bigcup\mathcal{U}_n$, there is a sequence $(U_n)_{n\in\mathbb{N}}$ such that $U_n \in \mathcal{U}_n$ for each $n \in \mathbb{N}$ and $A \subset \bigcup_{n\in\mathbb{N}}\mathsf{St}(U_n,\mathcal{U}_n)$ (respectively, $A \subset \overline{\bigcup_{n\in\mathbb{N}}\mathsf{St}(U_n,\mathcal{U}_n)}$, $A \subset \bigcup_{n\in\mathbb{N}}\overline{\mathsf{St}(U_n,\mathcal{U}_n)}$, $A \subset \bigcup_{n\in\mathbb{N}}\mathsf{St}(\overline{U_n},\mathcal{U}_n))$;*

(4) *$\mathcal{C}$-strongly star Rothberger (respectively, weakly $\mathcal{C}$-strongly star Rothberger, almost $\mathcal{C}$-strongly star Rothberger) if, for each $A \in \mathcal{C}$ and each sequence $(\mathcal{U}_n)_{n\in\mathbb{N}}$ of collections of sets open in $X$ such that $\overline{A} \subset \bigcup\mathcal{U}_n$, there is a sequence $(x_n)_{n\in\mathbb{N}}$ of elements of $\overline{A}$ such that $A \subset \bigcup_{n\in\mathbb{N}}\mathsf{St}(x_n,\mathcal{U}_n)$ (respectively, $A \subset \overline{\bigcup_{n\in\mathbb{N}}\mathsf{St}(x_n,\mathcal{U}_n)}$, $A \subset \bigcup_{n\in\mathbb{N}}\overline{\mathsf{St}(x_n,\mathcal{U}_n)})$;*

(5) *$\mathcal{C}$-star Hurewicz (respectively, almost $\mathcal{C}$-star Hurewicz, faintly $\mathcal{C}$-star Hurewicz) if, for each $A \in \mathcal{C}$ and each sequence $(\mathcal{U}_n)_{n\in\mathbb{N}}$ of collections of sets open in $X$ such that $\overline{A} \subset \bigcup\mathcal{U}_n$, there is a sequence $(\mathcal{V}_n)_{n\in\mathbb{N}}$ such that $\mathcal{V}_n$ is a finite subset of $\mathcal{U}_n$ for each $n \in \mathbb{N}$ and each $x \in A$ belongs to all but finitely many sets $\mathsf{St}(\bigcup\mathcal{V}_n,\mathcal{U}_n)$ (respectively, to all but finitely many sets $\overline{\mathsf{St}(\bigcup\mathcal{V}_n,\mathcal{U}_n)}$ to all but finitely many $\mathsf{St}(\overline{\bigcup\mathcal{V}_n},\mathcal{U}_n))$;*

(6) *$\mathcal{C}$-strongly star Hurewicz (respectively, almost $\mathcal{C}$-strongly star Hurewicz) if, for each $A \in \mathcal{C}$ and each sequence $(\mathcal{U}_n)_{n\in\mathbb{N}}$ of collections of sets open in $X$ such that $\overline{A} \subset \bigcup\mathcal{U}_n$, there is a sequence $(\mathcal{F}_n)_{n\in\mathbb{N}}$ of finite subsets of $\overline{A}$ such that each $x \in A$ belongs to all but finitely many sets $\mathsf{St}(F_n,\mathcal{U}_n)$ (respectively, to all but finitely many sets $\overline{\mathsf{St}(F_n,\mathcal{U}_n)})$.*

When $\mathcal{C}$ is the family of *all* nonempty subsets of $X$, then we say that $X$ is *set star Menger* (set-SM) (*set star Rothberger* (set-SR), *set star Hurewicz* (set-SH)), and similarly for other classes defined in the above definition.

Evidently, if a space $X$ belongs to $\mathcal{C}$ and $X$ is $\mathcal{C}$-set (strongly) star Menger, then $X$ is (strongly) star Menger.

Let set-SC and set SSC be abbreviations for set starcompact and set strongly starcompact spaces (for starcompact and strongly starcompact spaces, see [2,3]). Then, we have the following diagram for the Menger-type properties (see [91]).

$$\text{set}-\text{SSC} \rightarrow \text{set}-\text{SC}$$
$$\downarrow \qquad\qquad \downarrow$$
$$\text{Menger} \rightarrow \text{set}-\text{SSM} \quad\rightarrow\quad \text{set}-\text{SM}$$
$$\downarrow \qquad\qquad \downarrow$$
$$\text{SSM} \quad\rightarrow\quad \text{SM}$$

Diagram 1

We have the diagram for Hurewicz-type properties similar to Diagram 1 [95].

$$\text{set}-\text{SSC} \rightarrow \text{set}-\text{SC}$$
$$\downarrow \qquad\qquad \downarrow$$
$$\text{Hurewicz} \rightarrow \text{set}-\text{SSH} \quad\rightarrow\quad \text{set}-\text{SH}$$
$$\downarrow \qquad\qquad \downarrow$$
$$\text{SSH} \quad\rightarrow\quad \text{SH}$$

Diagram 2

Observe that there are examples showing that none of the implications in Diagrams 1 and 2 are reversible [92,95].

For example, the ordinal space $[0, \omega_1)$ with the order topology is a set strongly star Menger space (hence set star Menger) which is not Menger because it is not Lindelöf [1].

We give one more example.

**Example 1** ([92], Example 5). *There exists a $T_1$ set star-Menger space which is not set strongly star-Menger.*

Let $X = A \cup B$, where $A = \{a_\alpha : \alpha < \mathfrak{c}\}$ is any set with $|A| = \mathfrak{c}$ and $B$ is any countable set such that any element of $B$ is not in $A$. Topologize $X$ as follows:

(i) For each $a_\alpha \in A$ and each finite subset $F \subset B$, $\{a_\alpha\} \cup (B \setminus F)$ is a basic open neighborhood of $a_\alpha$;

(ii) Each element of $B$ is isolated.

**Example 2** ([95], Example 2.4). *There exists a Hausdorff star Hurewicz space, which is not set star Hurewicz.*

In some classes of spaces, certain properties from Diagram 1 coincide.

**Theorem 17.** *If $X$ is a paracompact Hausdorff space, then the following are equivalent:*

(1) *$X$ is Menger;*
(2) *$X$ is set strongly star-Menger;*
(3) *$X$ is strongly star-Menger;*
(4) *$X$ is set star-Menger;*
(5) *$X$ is star-Menger.*

The subspace $Y = \{\alpha + 1 : \alpha \text{ is a limit ordinal}\}$ of $[0, \omega_1)$ that is not set star Menger (hence not set strongly star Menger). Therefore, the properties set-SM and set-SSM are not hereditary. However, set star-Mengerness and set strongly star-Mengerness are preserved by clopen subspaces. They are also preserved by continuous mappings.

What about preimages? We give a result on preimages of set strongly star-Mengerness. For this we need a new concept defined as follows. We call a space *X* *nearly set stongly star-Menger* if, for each $A \subset X$ and each sequence $(\mathcal{U}_n : n \in \mathbb{N})$ of open covers of *X*, there is a sequence $(F_n : n \in \mathbb{N})$ of finite subsets of *X* such that $A \subset \bigcup_{n \in \mathbb{N}} \mathrm{St}(F_n, \mathcal{U}_n)$.

**Theorem 18** ([92], Theorem 3.5). *Let $f : X \to Y$ be an open and closed, finite-to-one continuous mapping from a space X onto a set strongly star-Menger space Y. Then, X is nearly set strongly star-Menger.*

The product of two set star-Menger spaces need not be set star-Menger. In fact, there exist two countably compact spaces *X* and *Y* such that $X \times Y$ is not set star-Menger.

Moreover, there exist a set star-Menger space *X* and a Lindelöf space *Y* such that $X \times Y$ is not set star-Menger.

There exist a countably compact (hence, set star-Menger) space *X* and a Lindelöf space *Y* such that $X \times Y$ is not set star-Menger.

Indeed, let $X = [0, \omega_1)$ be equipped with the order topology, and let $Y = [0, \omega_1]$ with the following topology. Each point $\alpha < \omega_1$ is isolated, and a set *U* containing $\omega_1$ is open if and only if $Y \setminus U$ is countable. Then, *X* is countably compact (hence, set star-Menger) and *Y* is Lindelöf. However, $X \times Y$ is not set star-Menger.

For more results on set star selection principles, we refer to the papers [31,96–98] and the very recent article [81].

## 5. Directions of Further Investigation

We end this chapter by proposing some lines of the future research in the field of star selection properties. We suggest the investigation of the properties defined below.

### 5.1. Related to the Classical Star Selection Principles

**Definition 11.** *Let p be a given natural number, and let $\mathcal{A}$ and $\mathcal{B}$ be collections of some kind of open covers of a space X. X is said to satisfy:*

*(1) $\mathsf{S}_\mathsf{p}^*(\mathcal{A}, \mathcal{B})$ (respectively, $\mathsf{SS}_\mathsf{p}^*(\mathcal{A}, \mathcal{B})$) if, for each sequence $(\mathcal{U}_n : n \in \mathbb{N})$ of elements of $\mathcal{A}$, there is a sequence $(\mathcal{V}_n : n \in \mathbb{N})$ (respectively, a sequence $(F_n : n \in \mathbb{N})$) such that for each n, $\mathcal{V}_n \in [\mathcal{U}]^{\leq p}$ (respectively, $F_n \in [X]^{\leq p}$) and $\{\mathrm{St}(\bigcup \mathcal{V}_n, \mathcal{U}_n) : n \in \mathbb{N}\} \in \mathcal{B}$ (respectively, $\{\mathrm{St}(F_n, \mathcal{U}_n) : n \in \mathbb{N}\} \in \mathcal{B}$).*

- $\mathsf{S}_\mathsf{p}^*(\mathcal{O}, \mathcal{O})$ *is called p-star-Menger* $\mathsf{SM}_\mathsf{p}$;
- $\mathsf{SS}_\mathsf{p}^*(\mathcal{O}, \mathcal{O})$ *is called p-strongly star Menger* $\mathsf{SSM}_\mathsf{p}$;
- $\mathsf{S}_\mathsf{p}^*(\mathcal{O}, \Omega)$ *is called p-$\omega$-star-Menger* $\omega\mathsf{SM}_\mathsf{p}$;
- $\mathsf{SS}_\mathsf{p}^*(\mathcal{O}, \Omega)$ *is called p-$\omega$-strongly star Menger* $\omega\mathsf{SSM}_\mathsf{p}$;
- $\mathsf{S}_\mathsf{p}^*(\mathcal{O}, \Gamma)$ *is called p-star-Hurewicz* $\mathsf{SH}_\mathsf{p}$;
- $\mathsf{SS}_\mathsf{p}^*(\mathcal{O}, \Gamma)$ *is called p-strongly star Hurewicz* $\mathsf{SSH}_\mathsf{p}$.

Notice that for $p = 1$, we have the classical star Rothberger-type properties.

It would be interesting to also study spaces which are $\mathsf{S}_\mathsf{p}^*(\mathcal{A}, \mathcal{B})$ and $\mathsf{SS}_\mathsf{p}^*(\mathcal{A}, \mathcal{B})$ for *some* $p \in \mathbb{N}$.

Additionally, the situation when in the above definition for each n, $|\mathcal{V}_n| \leq n$ or $|F_n| \leq n$ is worthwhile to investigate.

### 5.2. Related to Section 3

**Definition 12** ([85]). *A topological space X is said to be:*

- Weakly M-acc *(shortly,* wM-acc*) (respectively,* almost M-acc *(shortly,* aM-acc*)) if, for each sequence $(\mathcal{U}_n : n \in \mathbb{N})$ of open covers of X and each sequence $(D_n : n \in \mathbb{N})$ of dense subsets of X, there are finite sets $F_n \subset D_n$, $n \in \mathbb{N}$, such that $\overline{\bigcup_{n \in \mathbb{N}} \mathrm{St}(F_n, \mathcal{U}_n)} = X$ (respectively, $\bigcup_{n \in \mathbb{N}} \mathrm{St}(F_n, \mathcal{U}_n) = X$).*

- Weakly R-acc *(shortly,* wR-acc*) (respectively,* almost R-acc *(shortly,* aR-acc*)) if, for each sequence* $(\mathcal{U}_n : n \in \mathbb{N})$ *of open covers of X and each sequence* $(D_n : n \in \mathbb{N})$ *of dense subsets of X, there are* $a_n \in D_n$, $n \in \mathbb{N}$ *such that* $\bigcup_{n \in \mathbb{N}} \mathrm{St}(a_n, \mathcal{U}_n) = X$ *(respectively,* $\bigcup_{n \in \mathbb{N}} \overline{\mathrm{St}(a_n, \mathcal{U}_n)} = X$*).*
- almost H-acc *(shortly,* aH-acc*) if, for each sequence* $(\mathcal{U}_n : n \in \mathbb{N})$ *of open covers of X and each sequence* $(D_n : n \in \mathbb{N})$ *of dense subsets of X, there are finite sets* $F_n \subset D_n$, $n \in \mathbb{N}$ *such that each* $x \in X$ *belongs to* $\overline{\mathrm{St}(F_n, \mathcal{U}_n)}$ *for all but finitely many n.*

Parallel to the study of properties of classes of spaces defined in the previous definition, it may be interesting to find examples distinguishing these properties from the properties M-acc, R-acc, and H-acc.

**Funding:** This research received no external funding.

**Data Availability Statement:** Not applicable.

**Acknowledgments:** The author is grateful to the three anonymous referees for their careful reading of the paper and useful comments and suggestions, which led to a number of improvements in the exposition.

**Conflicts of Interest:** The author declares no conflict of interest.

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
