# Peer review of "On Star Selection Principles Theory"

_axioms, doi:10.3390/axioms12010093_

Round 1

Reviewer 1 Report

    In this paper, the author reviewed up-to-date recent results in the field of star selection principles, rapidly growing area of topology, and to present a few new results.

    My comments on this paper are as follows:

  ∙  The design and language of this paper are very good and acceptable .

  ∙  In the first section, the history of this problem was mentioned and the aim of this paper was introduced.

  ∙  In second section, some necessary concepts related to star selection principles and hyperspaces were recalled.

  ∙  In third section, selective version of the acc and (a) properties were given.

  ∙  In fourth section, set star selection properties were given.

  ∙  In the last section, Directions of further investigation were given.

    In my opinion, this paper will be useful for star selection principle and general topology.

    After the above comments, my decision on this paper is ACCEPT.

Author Response

Dear Referee,

Thank you very much for your positive comments on my article. 

Reviewer 2 Report

I recommend this paper for publication after the items in the attached document are addressed.

Author Response

Dear Referee,

Thank you very much for your veey useful and valuable comments abd suggestions. In the attached file you will find changes done nby me (following all your remarks).

PS. Unfortunatwly, I cannot obtain the pdf file of my revision (it will be done by editors of Axioms)

Best regards

Ljubisa Kocinac

Reviewer 3 Report

See my PDF file!

Author Response

Dear Referee, 

Thank you very much for your useful comments and suggestions. 

I revised the paper following your instuctions (see the attached file which is my response to your report).

PS. Unfortunately, I cannot obtain the pdf file of the paper. It will be done by the editors of Axioms. My changes are in the Latex file which will be sent to the editors.

Best regards

Ljubisa Kocinac
